# Estimation of the Adhesion Interface Performance in Aluminum-PLA Joints by Thermographic Monitoring of the Material Extrusion Process

**DOI:** 10.3390/ma13153371

**Published:** 2020-07-29

**Authors:** Stephan Bechtel, Mirko Meisberger, Samuel Klein, Tobias Heib, Steven Quirin, Hans-Georg Herrmann

**Affiliations:** 1Chair for Lightweight Systems, Saarland University, Campus E3 1, 66123 Saarbrücken, Germany; Mirko.Meisberger@web.de (M.M.); Samuel.Klein@izfp.fraunhofer.de (S.K.); tobias.heib@izfp-extern.fraunhofer.de (T.H.); steven.quirin@uni-saarland.de (S.Q.); hans-georg.herrmann@izfp.fraunhofer.de (H.-G.H.); 2Fraunhofer Institute for Nondestructive Testing IZFP, Campus E3 1, 66123 Saarbrücken, Germany

**Keywords:** additive manufacturing, material extrusion, thermographic process monitoring, polymer–metal hybrid structures, adhesion interface performance

## Abstract

Using additive manufacturing to generate a polymer–metal structure offers the potential to achieve a complex customized polymer structure joined to a metal base of high stiffness and strength. A tool to evaluate the generated interface during the process is of fundamental interest, as the sequential deposition of the polymer as well as temperature gradients within the substrate lead to local variations in adhesion depending on the local processing conditions. On preheated aluminum substrates, 0.3 and 0.6 mm high traces of polylactic acid (PLA) were deposited. Based on differential scanning calorimetry (DSC) and rheometry measurements, the substrate temperature was varied in between 150 and 200 °C to identify an optimized manufacturing process. Decreasing the layer height and increasing the substrate temperature promoted wetting and improved the adhesion interface performance as measured in a single lap shear test (up to 7 MPa). Thermographic monitoring was conducted at an angle of 25° with respect to the substrate surface and allowed a thermal evaluation of the process at any position on the substrate. Based on the thermographic information acquired during the first second after extrusion and the preset shape of the polymer trace, the resulting wetting and shear strength were estimated.

## 1. Introduction

In recent years, additive manufacturing (AM) technologies have gained importance in product development, prototyping and manufacturing for small series. These technologies offer huge potentials not only technically, but also in terms of sustainable and customized solutions [1,2]. In the case of the widespread material extrusion (ME), the material, usually a polymer, is fed through an extrusion nozzle and dosed in a targeted way—cf. DIN EN ISO/ASTM 52900-2017. By rastering the extrusion head, a 3 dimensional part is generated layer-by-layer. This process is often referred to as fused deposition modeling^®^ (FDM^®^) or fused filament fabrication (FFF). For many applications, the demands, including functionality and costs, vary locally in a part or structure. Multi-material parts are taken into account to address these demands by varying the material as well as the fabrication process (cf., e.g., [3]). For example, our recent studies within the DFG-founded research program SPP1712 have shown that a thermoplastic interlayer can improve the mechanical performance in the cyclic loading regime of a hybrid-structure—a metal insert embedded in a carbon fiber-reinforced polymer (CFRP) laminate—(cf., e.g., [4]). However, the interfaces represent damage-critical areas in the multi-material compound structure. Moreover, in the case of ME, the sequential deposition of the thermoplastic polymer onto a metal substrate leads to local variations in adhesion depending on the local processing conditions.

Spoerk et al. [5,6] evaluated the adhesion interface performance of several extruded thermoplastics onto different non-metallic substrates using an in-house built adhesion measurement device. Increasing the printing bed temperature above the glass transition temperature, T_g_, of the polymer significantly increases the adhesion forces and prevents the part from detaching from the printing bed during the build process.

From adhesive technology, the adhesion interface performance of hotmelts is known to be highly sensitive to the local temperature management during the application of the adhesive. Due to the high thermal conductivity of metal substrates, Habenicht et al. [7,8] suggests preheating the substrates to the melting temperature, T_m_, of the hotmelt prior to its application. Decreasing the viscosity of the adhesive promotes wetting of the adherent, which is an indispensable precondition for structural polymer–metal bonding.

By processing thermoplastics through injection molding, the bonding of the polymer to a metal insert can be improved via local temperature management in the polymer metal contact region (cf., e.g., [9,10,11]). Furthermore, the injection and holding pressure allows the formation of a forced polymer–metal bonding via penetration of the polymer in the surface textures, which is often referred to as polymer to metal direct adhesion (cf., e.g., [9,12]). However, in the case of the ME process with a conventional setup, no pressure can be applied after the polymer has left the extrusion nozzle.

For the most common thermoplastics currently used in the ME process, Amancio-Filho et al. [13] suggests substrate temperatures in between the extrusion, T_e_, and the crystallization temperature, T_c_, of the polymer to optimize the polymer–metal bonding. In their recent publications regarding the Addjoining^®^ process, Falck et al. [14,15] applied a primer by spreading, e.g., an acrylonitrile butadiene styrene (ABS)-acetone solution onto the metal surface before the actual ME process is carried out. An improvement in single lap shear strength was observed for an increased extruder head velocity, primer thickness and decreased layer height [14]. Chueh et al. [16] generated, e.g., polylactic acid (PLA)-steel joints by ME of the polymer on a preheated (180 °C) structured metal substrate with undercuts. This resulted in a combination of a form-fit and an adhesive connection. Hertle et al. [17] observed an increase in lap shear strength of ME joined polypropylene (PP)-aluminum samples by increasing the substrate and extrusion temperature, respectively. The increased contact temperature resulted in an improved filling of the microstructures of the electrochemically treated aluminum surface. While Herlte et al. [17] observed a large influence of the substrate temperature, T_s_, Falck et al. [14] reported only a minor effect. This difference is attributed to the fact, that Falck et al. applied a primer beforehand. Hence, during the ME process, the polymer is not deposited directly on the metal surface. The significantly lower thermal conductivity of the primer reduces the effect of the substrate temperature (cf., e.g., [8]).

An optimized local temperature management allows for an improvement of the adhesive (polymer layer to substrate) and cohesive (layer to layer) properties [5,6,7,13,14,15,17,18,19,20]. Thermography is suitable to evaluate the ME process in terms of local temperature distribution [18,19,21,22]. In contrast to contact thermometers, the thermal radiation based temperature measurement method allows quickly changing temperature distributions on the surface of objects with low thermal conductivity and heat capacity to be accessed without contact [23]. For an accurate temperature determination, the local emissivity must be taken into account [23,24].

Sepalla et al. [21] characterized the weld formation between two ABS layers by means of thermographic process monitoring. The weld exhibits cooling rates of up to 100 K/s and stays above the glass transition temperature, T_g_, for approximately 1 s. Furthermore, based on temperature dependent polymer dynamic processes (rheology), Sepalla et al. [19] correlated the local temperature management with the mechanical interlayer strength (Mode III fracture). By increasing the effective weld time, the bonding can be improved up to 70% of the bulk strength.

Bartolai et al. [25] predicted the strength of ABS and PC samples produced by ME using a polymer weld theory based on rheological and thermographic measurements.

This article aims to investigate the ability of thermographic process monitoring to estimate the resulting structural and mechanical properties in the polymer–metal contact region. A tool to evaluate the local structure at the interface during the process is of fundamental interest, as the sequential deposition of the polymer, as well as temperature gradients within the substrate lead to time-dependent local variations in wetting and adhesion depending on the local processing conditions. The process parameters substrate temperature, T_s_, and layer height, d_Po_, which were varied in this study, are often adjusted depending on the use case and are expected to affect the adhesion interface performance (cf., e.g., [8,14,17]).

A model system compromising of a technical polylactic acid (PLA) and aluminum 6082 was examined. Based on the caloric and thermo-rheological properties of the polymer, which were obtained by differential scanning calorimetry (DSC) and rheometry, a relevant range of process parameters in terms of extrusion, T_e_, and substrate temperature, T_s_, were chosen. The local temperature during the deposition process was examined with infrared thermography. Optical microscopy was used to evaluate the structure in the polymer–metal contact region in terms of wetting behavior via contact angle measurement. Finally, the resulting mechanical performance was evaluated based on single lap shear tests.

## 2. Materials and Methods 

### 2.1. Materials and Sample Preparation

#### 2.1.1. Aluminum Substrates

The aluminum substrates were received from the water jet cutting company RS-Evolution (66793 Saarwellingen, Germany). The deburred substrates were prepared to measure 25 mm by 115 mm from a 2 mm thick sheet metal of EN AW-6082-T6. This medium strength aluminum alloy has excellent corrosion resistance and is typically used for structural parts in, e.g., the transportation sector [26]. The substrates were sandblasted with corundum (Al_2_O_3_) of the size F150, which is received from Oberflächentechnik Seelmann (06847 Dessau-Roßlau, Germany). The particle size was about 82 µm and lay within the standardized range (45–106 µm)—cf. DIN EN 13887-2003. Sandblasting was performed with a ST 800-J Auer Strahltechnik (68309 Mannheim, Germany) at a pressure of 6 bar, a working distance of 10 cm and an angle of 90° to the surface.

#### 2.1.2. Polylactic Acid (PLA)

PLA is one of the most extensively researched and used biobased and renewable polymers. The aliphatic polyester can be made from annually renewable resources and has the potential to replace conventional petrochemical-based polymers for many industrial applications [27]. The Biopolymer Ingeo™ 3D870 (Nature Works, Minnetonka, MN, USA), purchased as a black colored filament, was used as a polymer component.

The polymer was processed with a customized ME machine based on a desktop FFF platform (Ender 3, Creality 3D Technology Co., Ltd., Shenzhen, China). The metal substrates were fixed on an aluminum hot plate and were preheated to a temperature, T_s_, ranging from 150 to 200 °C. The extruder was equipped with a water-cooled heatsink, a volcano hotend and a brass nozzle with a diameter, w_Po_, of 0.8 mm (all purchased from E3D Online, Oxfordshire, UK). The extruder temperature, T_e_, was set to 200 °C. The extruder and bed temperature were chosen based on the thermal and thermo-rheological properties of the PLA. Figure 1 shows the temperature profile during the production of a single lap joint (SLJ) specimen, in the case of a substrate temperature, T_s_, of 200 °C. For the SLJ specimens, the substrate temperature, T_s_, was gradually decreased after each layer. Independent of T_s_ for the first layer, the substrate temperature was decreased to 150 °C, 100 °C and 60 °C for the second, third and the remaining layers, respectively. During the deposition of the d_Po_ = 0.3 mm high layers, the extruder moved at a velocity, v_e_, of 10 mm/s. For the first layer, d_Po_ was varied between 0.3 and 0.6 mm, while v_e_ stayed at 10 mm/s. For the mechanical test specimens, the extruded tracks were oriented parallel to the loading direction.

The extruder, T_e_, and substrate temperature, T_s_, were controlled based on thermocouples placed in the hotend and hot plate, respectively (cf. Figure 2). For sample preparation, three samples were made per sequence on the identical substrates A, B and C (cf. Figure 2). Since the polymer–metal interaction was of particular interest in this study, thermographic process monitoring is only discussed for the first layer of polymer.

### 2.2. Methods

#### 2.2.1. Thermography (Thermal Process Monitoring)

Thermography was performed in the spectral range between 7.5 and 14 µm with an InfraTec (01217 Dresden, Germany) VarioCAM HD head equipped with a 30 mm objective and a close-Up 0.5 × macro. Acquisition was realized with the Software IRBIS 3.0 at a frequency of 30 Hz. The maximal spatial resolution and the noise equivalent temperature deviation (NETD) were given with 32 µm and 0.05 K, respectively. The measuring uncertainty of the temperature was up to 1.5%. The working distance was 9 cm and the measurement setup was according to Figure 2. Data processing was carried out with MATLAB R2019a (MathWorks).

Three different measuring configurations with respect to the viewing angle, Φ, which is defined as the angle between the substrate surface and the optical axis, can be distinguished. The resulting polymer surface temperature is termed as T_IR, Φ=0°_, T_IR, Φ=25°_ and T_IR, Φ=90°_. Each of those configurations has its advantages and disadvantages. On the one hand, measuring T_IR, Φ=0°_ allows the determination of temperature distributions in the vertical (x–z) plane. To eliminate shadowing generated by the metal substrate, the polymer must be extruded right at the front edge of the substrate. Hence, the generation of a two dimensional polymer–metal bond, i.e., the SLJ specimen, cannot be monitored. On the other hand, measuring T_IR, Φ=90°_ allows the determination of temperature distributions in the horizontal (x–y) plane. However, with the IR-camera in use, only ex-situ measurements were possible, as the extruder blocks the view during the extrusion process. In order to monitor the material extrusion at any position on the substrate surface, it was intended to determine the polymer surface temperature by measuring at a viewing angle, Φ, of 25°.

The thermal radiation based temperature measurement method was used to access quickly changing temperature distributions on the surface of the polymer, which possesses a low thermal conductivity (cf. Table 1). The local spectral radiance of a black body, $L_{\lambda ,S}\left(T\left(x,y\right)\right)$, depends on the local temperature, T(x,y), and the wavelength, λ, with applicable first and second radiation constant, c_1L_ and c_2_, according to Plank’s law (cf. Equation (1)). For a real body, the spectral radiance, $L_{\lambda }\left(\lambda ,T,\theta \right)$, is reduced, due to the angular (θ), wavelength (λ) and temperature (T) dependent emissivity, ε. For the thermographic temperature measurement with a broadband detector, the radiance detected, L_D_, can be approximated according to Equation (2) within the relevant spectral range between λ_1_ = 7.5 and λ_2_ = 14 µm. It mainly depends on the polymer temperature, T_Po_, and the temperature of the surrounding, T_Sur_, if the emissivity of the polymer, $\varepsilon_{\lambda }\left(T_{Po},\theta \right)$, was measured and averaged in the considered conditions.
(1)$$L_{\lambda ,S}\left(T\left(x,y\right)\right)=\frac{c_{1L}}{\lambda^{5}}\cdot {\left(\exp\left[\frac{c_{2}}{\lambda \cdot T\left(x,y\right)}\right]-1\right)}^{-1}$$
(2)$$\int^{\;}L_{D}\left(\lambda ,T_{Po},T_{sur},\theta \right)={\bar{\varepsilon }}_{\lambda }\left(T_{Po},\theta \right)\cdot \int_{\lambda_{1}}^{\lambda_{2}}L_{S}\left(\lambda ,T_{Po}\right)\;d\lambda +\left(1-{\bar{\varepsilon }}_{\lambda }\left(T_{Po},\theta \right)\right)\cdot \int_{\lambda_{1}}^{\lambda_{2}}L_{S}\left(\lambda ,T_{sur}\right)\;d\lambda$$

#### 2.2.2. Differential Scanning Calorimetry (Caloric Properties)

DSC was performed with a TA Instruments (New Castle, DE, USA) Q100. The measuring cell was nitrogen-purged (20 ml/min). About 4 mg of extruded PLA were placed in an aluminum crucible, vapor-plated with gold. The same type of crucible was taken as reference.

DSC was used to access the glass transition, crystallization and melting events. Heating and cooling was realized at a rate of 10 K/min to 230 °C with a subsequent isothermal homogenization for 10 min. This rate was similar to the cooling rate of the metal substrates after the joining process (cf. Figure 1).

#### 2.2.3. Rheometry (Thermo-Rheological Properties)

Rheometry was performed with a TA Instrument (New Castle, DE, USA) Ar2000ex using the plate–plate configuration. A 1 mm thick PLA disc with a diameter of 25 mm was generated by ME and was placed in between the plates of the rheometer. After preheating to 200 °C, the PLA disc was carefully compressed by the rheometer plates to prevent squeezing out the polymer. Using the method of deformation-controlled oscillation rheology, the deformation amplitude was set to 5%. After equalization for 2 min at the set measurement temperature, which ranged from 200 to 120 °C, oscillation sweeps were carried out in the range from ω = 2π·10 to ω = 2π·0.01 Hz.

Rheometry was used to access the thermo-rheological properties of the polymer, which are crucial for wetting, cohesive properties [28,29,30] and adhesion interface performance [7].

#### 2.2.4. Light Microscopy (Wetting Behavior)

Brightfield microscopy was performed with a Leica (35578 Wetzlar, Germany) DM6000 Multifocus. To assess the wetting behavior, single tracks of the polymer were extruded on the preheated substrates. After cooling, a perpendicular cut (Steuers Discotom 6, cutting speed 0.1 mm/s) through those contact angle measurement (CAM) specimens was examined by light microscopy. In those brightfield images, the contact angle between the polymer and the metal part was measured with the software Image Access Premium (Imagic Bildverarbeitung AG, Glattbrugg, Switzerland) to evaluate the wetting behavior.

#### 2.2.5. Tensile Tester (Mechanical Performance)

Tensile tests were carried out with an Instron 8500 equipped with a 100 kN loadcell and a self-made tempering cell to ensure reproducible test conditions by means of thermal stability at 23 °C. The adhesion interface performance in PLA-aluminum assemblies was evaluated based on ISO 19095 (Type B, without specimen retainer). Deviating from the standard, the joint area was increased from 5 mm × 10 mm to 10 mm × 20 mm, to reduce edge effects and improve handling. Furthermore, the length of the aluminum lap was increased to 115 mm to allow an in-situ ultrasonic (SH-waves) monitoring of the joint failure (not included in this article). The sample dimensions are shown in Figure 3. The tensile tests were driven displacement controlled with 2 mm/min cross-head speed. The tensile single-lap-shear strength, $\tau_{SLJ}=F_{B}/A_{J}$, was calculated based on the breaking load, F_B_, and the joint area, A_J_. Tensile tests were carried out within 5 h after sample production to reduce aging effects.

## 3. Results and Discussion

### 3.1. Material Properties

Table 1 gives a summary of the relevant properties of the used aluminum 6082-T6 substrates and PLA Ingeo™ 3D870. The properties of the technical polymer component depend on the additives included, as well as on the degree of polymerization. The mechanical performance of the additively produced structure depends on a variety of process and post-processing parameters.

The caloric and thermo-rheological properties of the polymer are highly relevant to conceive the fundamentals of the process-structure-properties relationship. Figure 4 depicts the heat flow as a function of temperature as obtained by DSC.

The temperature ranges for the glass transition, T_g_ = 60.5 °C, post-crystallization, T_pc_ = 95.1 °C, melting, T_m_ = 175.2 °C and crystallization, T_c_ = 114.1 °C (cf. Figure 4), as obtained in the first DSC cycle, are compliant with data from datasheet [32] and literature (cf., e.g., [27,33]).

The temperature and shear rate-dependent viscosity was determined by rheology. According to the Cox-Merz rule, $\left|\eta^{*}\left(\omega \right)\right|=\eta \left(\dot{\gamma }=\omega \right)$, which applies to PLA [34], the steady shear viscosity is plotted as a function of temperature for three different shear rates in Figure 5. With increasing temperature and shear rate, the viscosity decreased. The absolute values are comparable with those reported by Benwood et al. [35]. At about 140 °C, there was a distinctive change in viscosity, which can be related to crystallization processes. This is in good agreement with the caloric-determined temperature range for the crystallization (cf. Figure 4). This range started at about 120 °C, but depended on the cooling rate. Within the rheometer, the undefined cooling rate was considerably below 10 K/min; therefore, the crystallization processes could also occur at 140 °C. Above 140 °C, shear thinning becomes more pronounced with increasing shear rate. This “Carreau fluid” behavior (cf., e.g., [36]) was also observed by Benwood et al. [35].

For the application of hotmelts, Habenicht et al. [8] reports characteristic processing viscosities in the range of 20 and 10^4^ Pa·s depending on the type of adhesive and the use case. In particular, he suggests preheating of the metal substrates above the melting temperature of the hotmelt (T_m,PLA_ = 176 °C). The typical processing range in terms of viscosity for extrusion is 10^3^–10^4^ Pa·s (cf., e.g., [36]). Increasing the temperature decreases the viscosity and promotes wetting; however, this also amplifies thermal degradation (cf., e.g., [37]).

For the ME process, the extrusion temperature, T_e_, was set to a nominal value of 200 °C. This temperature is within the suggested range of the filament supplier (190–230 °C). The resulting viscosity fell within the lower portion of the typical range for extrusion (cf. Figure 5). The substrate temperature, T_s_, varied between 150 and 200 °C. The former corresponds to the upper viscosity limit suggested for processing of hot melts and extrusion (cf. Figure 5). The latter is equal to T_e_.

### 3.2. Thermal Process Characterization by Means of Thermography

First, the stability of the material extrusion process by means of temperature of the extruded polymer was investigated. The polymer was extruded at different rates and temperatures for 5 s. Thermography (Φ = 0°, Figure 2) was used to measure the temperature of the polymer right below the extrusion nozzle. According to Figure 6, for the highest observed extrusion rate (24 mm^3^/s), the apparent surface temperature of the extruded polymer, T_IR, Φ=0°, app_, significantly decreased within the first few seconds. Additionally, the extrusion rate of 12 mm^3^/s led to a slight decrease in T_IR, Φ=0°, app_ over time. For the volumetric extrusion rates of 2.4 and 4.8 mm^3^/s, T_IR, Φ=0°, app_ did not decrease over time. Hence, those extrusion rates were considered as thermally stable (note: a deviation of the internal polymer temperature was not taken into account). The thermally stable extrusion rates correspond to the used processing with an extruder head velocity of v_e_ = 10 mm/s and a layer height of d_Po_ = 0.3 mm and d_Po_ = 0.6 mm, respectively.

In order to estimate the emissivity of the polymer, the first two seconds of the extrusion process were taken into account. Here, T_IR, Φ=0°, app_ increased with increasing extrusion rate. For the lower extrusion rates (2.4–12 mm^3^/s), T_IR, Φ=0°, app_ was reduced at the beginning of extrusion, due to cooling of the polymer at the tip of the extrusion nozzle. For the same reason, the highest observed values of T_IR, Φ=0°,app_ were lower for the two lowest extrusion rates (2.4 and 4.8 mm^3^/s). The maximum in T_IR, Φ=0°, app_ was comparable (± 0.5 K, dashed line in Figure 6) for the two highest extrusion rates (12 and 24 mm^3^/s). Thus, this temperature was assumed to be equal to the set temperature of the extruder (T_e_ = 200 °C). An estimation for the emissivity was obtained by solving Equation (2) for ${\bar{\varepsilon }}_{\lambda }\left(T_{Po},\theta \right)$, where $\int^{\;}L_{D}\left(\lambda ,T_{Po},T_{sur},\theta \right)$ was the radiance detected by the camera, $\int_{\lambda_{1}}^{\lambda_{2}}L_{S}\left(\lambda ,T_{Po}\right)\;d\lambda$ and $\int_{\lambda_{1}}^{\lambda_{2}}L_{S}\left(\lambda ,T_{sur}\right)\;d\lambda$ were calculated based on Equation (1) for T_Po_ = 200 °C and T_sur_ = 20 °C. Hereafter, for the thermographic determination of the polymer surface temperature, T_IR_, the emissivity of the polymer was set to 0.78 and considered independent of its temperature and the observation angle.

Furthermore, the surface temperature of the substrate, T_IR,Φ=90°,s_, was compared to the surface temperature of the hot plate, T_IR,Φ=90°,hp_, which was controlled by a thermocouple (T_s_). For this purpose, the metal parts were covered with polyimide tape and the temperature distribution was determined by means of thermography (Φ = 90°, Figure 2). To account for local detachments between metal and tape, the maximum observed temperature within a measuring area of at least 1 cm^2^ × 1 cm² was used for the comparison. According to Figure 7, the temperature difference, T_IR,Φ=90°,hp_ − T_IR,Φ=90°,s_, increased with increasing set temperature and reached up to 6 K due to convective and radiative heat transfer. Moreover, temperature gradients within the substrates lead to variations of up to 5 K.

Hence, even substrates with simple geometry and high thermal conductivity show deviations from the set temperature as well as local variations in temperature. Therefore, monitoring of the local temperature during the sequential deposition of the polymer by ME is essential to evaluate the adhesion interface performance. This becomes even more relevant if the substrates exhibit an increased geometric complexity and a lower thermal conductivity.

#### Cooling Behavior of the Polymer During the ME Joining Process

The local polymer surface temperature, T_IR_(x,t), was computed as an average of three values right in the middle of the projected polymer track as displayed in the thermogram (cf. Figure 2). The T_IR_(t) signal was smoothed with a zero-phase moving average of 10 in order to reduce noise. This was especially essential before computing the cooling rate, T˙c=−∂T∂t. As indicated in Figure 8, the measuring configuration (cf. Figure 2), by means of the viewing angle (Φ = 0° vs. Φ = 25°) and position (substrate A vs. substrate B), had no significant influence on the observed polymer cooling behavior. Measuring T_IR, Φ=25°_ allows the ME process to be monitored at any position on the substrate surface hereafter. 

Figure 9 shows the timely change in T_IR, Φ=25°_ for a substrate temperature, T_s_, of 150, 180 and 200 °C for the two layer heights, d_Po_ = 0.3 and d_Po_ = 0.6 mm. In the moment of extrusion, the polymer was hottest and showed a temperature of about 200 °C, which conformed with the set extrusion temperature, T_e_. Within the next few seconds, the polymer cooled until it reached a plateau. The cooling was more pronounced and faster for lower substrate temperatures, T_s_. Furthermore, with decreasing layer height, d_Po_, the cooling rate increased significantly. The observed plateau temperatures, T_IR,plat_, were about 5–10 K below the set values for T_s_.

According to the difference in T_IR,plat_ between d_Po_ = 0.3 and d_Po_ = 0.6 mm the temperature gradient within the polymer was about 2 to 3 K per 0.3 mm (cf. Table 2). Considering T_s_ = 150 °C and d_Po_ = 0.3 mm, a reduction of the substrate surface temperature of about 3 K (cf. Figure 7) and a temperature gradient of 2 K per 0.3 mm in the polymer results in a polymer surface temperature of 145 °C, which is comparable to the thermographic determined polymer surface temperature (T_IR,plat_ = 143.8 ± 0.7 °C—cf. Table 2). Hence, the assumption of a temperature independent emissivity was appropriate for the investigated temperature range.

Table 2 gives an overview of average values for the thermographic determined extrusion, T_IR,e_, and plateau temperature, T_IR,plat_, as well as for the maximum cooling rate, ${\dot{T}}_{c,IR,max}$.

The observed extrusion temperature, T_IR, e_, increased with increasing substrate temperature, T_s_. This can be explained by a radiative and convective heating of the tip of the extrusion nozzle by the hot substrate. Decreasing the layer height, d_Po_, led to a slight increase in T_IR, Plat_ and a significant increase in ${\dot{T}}_{c,IR,max}$. The latter also increased significantly with decreasing substrate temperature, T_s_.

### 3.3. Wetting Behavior

Wetting is a necessary condition for adhesion [7]. In thermodynamic equilibrium, the wetting angle, φ, results from the interfacial tensions between polymer and substrate, γ_PS_, polymer and atmosphere, σ_PA_, and substrate and atmosphere, σ_SA_, according to Young’s equation (cf. Equation (3)).
(3)$$\cos\left(\varphi \right)=\frac{\sigma_{SA}-\gamma_{PS}}{\sigma_{PA}}$$

In general, clean metallic and oxidic surfaces possess a high surface tension, $\sigma_{SA}$, and are wetted properly by liquids with a low surface tension like polymers—$\sigma_{PA}$ [38]. However, Equation (3) does not account for differences between the advancing and receding angle, as well as for kinetically ruled process. In thermal equilibrium, liquids with an elevated viscosity, like polymer melts, wet the substrate gradually. The contact angle continuously decreases until it reaches equilibrium. Depending on the viscosity of the polymer melt, this process can take some 10 min (cf. e.g., [39]). In this study, the extruded polymer melt wetted the substrate while it cooled and, in the case of T_s_ = 150 °C, began to crystallize (cf. Figure 4 and Figure 5). Hence, the polymer underwent a rapid and distinctive change in viscosity. The resulting contact angle highly depended on the timely change in viscosity and corresponds to a frozen non-equilibrium state. Thus, a frozen state of the contact angle was achieved, which depends on the thermal history—namely the time at elevated temperature and cooling rate (cf. Figure 1). Therefore, the determined contact angles are only suited for a comparative assessment for the given setup and processing. The process parameters layer height, d_Po_, and width, w_Po_, determine the initial shape of the polymer trace. By assuming a cylinder segment geometry (cf. Figure 10), the initial contact angle, φ_cs,i_, can be calculated according to Equation (4).
(4)$$\cos\left(\varphi_{cs,i}\right)=\frac{r-d_{Po}}{r}=\frac{{\left(\frac{w_{Po}}{2}\right)}^{2}-d_{Po}{}^{2}}{{\left(\frac{w_{Po}}{2}\right)}^{2}+d_{Po}{}^{2}}\;\mathrm{With}\;r=\frac{{\left(\frac{w_{Po}}{2}\right)}^{2}+d_{Po}{}^{2}}{2d_{Po}}$$

Cross sections of the polymer metal contact region, as observed by light microscopy, are shown in Figure 11. The morphology of the polymer track was characterized by measures in contact angle, layer height and layer width. 

According to Figure 12, the contact angle decreased with increasing thermographic measured plateau temperature, T_IR, plat_. This is in conjunction with the temperature reaction of the viscosity (cf. Figure 5) and the general suggestions for the applications of hotmelts (cf. e.g., [7]). 

However, there is a large variation within each group. One reason may be found in the difference in time at an elevated temperature between the substrates A, B and C, which was up to 140 s and affected the gradual wetting (cf e.g., [39]). Additionally, the measured contact angles were always lower for the thinner layers. This observation is in conjunction with the preset shape of the polymer trace, which depends on the layer height, d_Po_, (cf. Equation (4) and Figure 10). Considering T_IR, Plat_ as well as the preset contact angle, φ_cs,i_, (cf. Equation (4)) allowed a superior estimation of the resulting contact angle, as shown in Figure 13. 

The plateau temperature, T_IR, plat_, which shows a clear correlation with the substrate temperature, T_s_, took up to 6 s to be reached (cf. Figure 9). However, during the ME process of an actual part it is usually not possible to monitor any point for such a long time with the current setup, as the extruder and substrate move, so the point of interest leaves either the field of view or the focus plane. Hence, it was attempted to correlate the plateau temperature, T_IR,plat_, with the thermographic information right after the extrusion. In order to define such a parameter, which strongly depends on T_IR,plat_ and is nearly independent of d_Po_, the cooling rate was considered in more detail.

The cooling process, T_IR_(t), can be approximated by an exponential law that has already been observed in early times as a result of the so called “Newton’s law of cooling“, which is best for conductive and convective, but not for radiative, heat transfer (cf. e.g., [40]). However, using this exponential decay function allowed an analytical estimation on how the cooling rate changes with time for different layer heights, d_Po_, and substrate temperatures, T_s_. Based on the assumption of uniform temperature distribution within the cross-sectional area of the polymer trace, semi-infinite length of the polymer trace and constant heat transfer coefficients, Bellehumeur et al. [41] give an analytical solution to describe the timely change in temperature of the deposited polymer. The model was adapted to the given geometry of the polymer trace (cf. Figure 10) and extended by distinguishing between heat transfer polymer–air and polymer–metal (cf. Equation (5)).
(5)$$T_{\mathrm{IR}}\left(t\right)=T_{\mathrm{IR},\mathrm{plat}.}+\left(T_{\mathrm{IR},e}-T_{\mathrm{IR},\mathrm{plat}.}\right)\cdot \exp\left(-\frac{t}{\tau_{c}}\right)\;\mathrm{With}\;\frac{1}{\tau_{c}}=v_{e}\;\frac{\sqrt{1+4\alpha \cdot \beta }-1}{2\alpha }\;\mathrm{Where}\;\alpha =\frac{k_{\mathrm{Po}}}{\rho_{\mathrm{Po}}\cdot c_{p,\mathrm{Po}}\cdot v_{e}}\;\mathrm{and}\;\beta =\frac{h_{\mathrm{Me}}\cdot P_{\mathrm{Me}}+h_{\mathrm{Air}}\cdot P_{\mathrm{Air}}}{\rho_{\mathrm{Po}}\cdot c_{p,\mathrm{Po}}\cdot A_{\mathrm{cs}}\cdot v_{e}}$$

The time constant, $\tau_{c}$, depends on thermal conductivity, k_Po_ = 0.15 W/m·K, heat capacity, c_p,Po_ = 2000 J/kg·K, density, ρ_Po_ = 1.24 g/cm³, of the polymer, heat transfer coefficient, h, extruder head velocity, v_e_, and the ratio of volume to surface area where the heat transfer takes place.

The heat transfer was a combination of a heat transfer polymer–air, $h_{Air}$, and a polymer–metal, $h_{Me}$, acting in parallel at the perimeters P_Air_ and P_Me_, respectively. For the polymer trace, approximated as a semi-infinite cylinder segment (cf. Figure 10), the cross-sectional area, A_cs_, and the perimeters, P_Air_ and P_Me_, can be calculated according to Equation (6).
(6)$$A_{\mathrm{cs}}=r^{2}\cdot \varphi_{\mathrm{cs},i}-\frac{w_{\mathrm{Po}}\cdot \left(r-d_{\mathrm{Po}}\right)}{2},\;P_{\mathrm{Air}}=2r\cdot \varphi_{\mathrm{cs},i}\;\mathrm{and}\;P_{\mathrm{Me}}=w_{\mathrm{Po}}$$

A good approximation of the experimental data was achieved with h_Air_ = 30 W/m²·K and h_Me_ = 870 W/m²·K for both layer heights (cf. Figure 14). Voids at the polymer–metal interface, as reported by Hertle et al. [17], reduce the heat transfer coefficient polymer–metal, h_Me_, and, ultimately, the thermographic measured maximum cooling rate, ${\dot{T}}_{c,IR,max}$, and plateau temperature, T_IR,plat_, of the polymer. For the analytical description, the polymer–metal interface was considered as homogeneous. The heat transfer coefficient h_Me_ was a combination of a heat transfer polymer–metal and a polymer–voids–metal. Thermographic monitoring did not reveal local differences in cooling behavior that would indicate the presence of large pores at the interface or within a polymer trace. This observation is supported by the cross sections of the polymer metal contact region, which revealed no large voids (cf. Figure 11).

Neglecting temperature dependence of material and heat transfer properties and, hence, of the time constant, $\tau_{c}$, the cooling rate at a fixed time, $t_{n}$, is proportional to the difference between initial, T_IR,e_, and plateau temperature, T_IR,plat._, (cf. Equation (7)).
(7)$$\dot{T}=\left(T_{IR,e}-T_{IR,plat.}\right)\cdot \exp\left(-\frac{t}{\tau_{c}}\right)\cdot \left(-\frac{1}{\tau_{c}}\right)\;\Leftrightarrow T_{IR,e}-T_{IR,plat.}=-\tau_{c}\cdot \exp\left(\frac{t}{\tau_{c}}\right)\cdot \dot{T}\;\overset{t=t_{n}}{\Rightarrow }\;T_{IR,e}-T_{IR,plat.}\;\propto \;\dot{T}\left(t=t_{n}\right)$$

For the given processing conditions in terms of layer height, d_Po_, and substrate temperature, T_s_, a comparable cooling rate, ${\dot{T}}_{c}$, for different layer heights could be found at about 0.5–1 s after extrusion (cf. marked range in Figure 15).

Hence, T_IR, plat_ was estimated based on ${\dot{T}}_{c,avg\left(0.5-1s\right)}$, which strongly depended on T_IR,plat_ and was nearly independent of d_Po_ (cf. Figure 16) ${\dot{T}}_{c,avg\left(0.5-1s\right)}$ is the average cooling rate within 0.5 and 1 s after extrusion.

.

The estimation of the wetting behavior in terms of the measured contact angle based on ${\dot{T}}_{c,avg\left(0.5-1s\right)}$ and φ_cs,i_ is shown in Figure 17.

### 3.4. Adhesion Interface Performance

The used single-lap configuration allowed a realizable sample preparation, as well as good accessibility for non-destructive testing and condition monitoring (not included in this article). The eccentric load path within the joint led to a rotation during loading and caused, among others, additional peel stresses (cf. e.g., [7]). Due to the different stiffness of the Al and the PLA part, the stress distribution in the bond line was asymmetric. Therefore, the determined single-lap-shear strength is rather suited for a comparative assessment than for a quantitative determination of the shear strength of the polymer–metal interface.

In analogy to the thermographic estimation of the contact angle (cf. Figure 17), Figure 18 shows the potential to estimate the single lap shear strength, τ_SLJ_, based on ${\dot{T}}_{c,avg\left(0.5-1s\right)}$ and φ_cs,i_. Differences between the monitoring of the CAM (isolated single tracks) and SLJ (adjoining tracks) specimens for the same processing conditions in terms of T_s_ and d_Po_ can be due to:
Different thermal boundary conditions exist due to the presence (SLJ) or absence (CAM) of a neighboring polymer track.The temperature measuring position on the polymer track might differ, as the boundaries of the polymer track were hard to segment in the case of the SLJ experiment.

The single lap shear strength, τ_SLJ_, increased with increasing substrate temperature, T_s_, and decreasing layer height, d_Po_ (cf. Figure 18), which is in conjunction with e.g., [8,14,17]. Additionally, the decrease in the average cooling rate, ${\dot{T}}_{c,avg\left(0.5-1s\right)}$, and preset contact angle, φ_cs,i_, also led to a change in failure mode from adhesive to cohesive failure. For improved wetting conditions, the shear strength of the PLA-Al interface exceeded the shear strength of the PLA-PLA interface. The variance in the experimentally determined τ_SLJ_ can be due to a couple of reasons:
The joint failure is very sensitive to local heterogeneities due to processing and substrate surface condition (e.g., roughness, contaminations and pores).Differences in the coefficient of thermal expansion (cf. Table 1) lead to internal stresses during cooling. Relaxation of the residual stresses, which can weaken or strengthen the bonding, is a time dependent process and depends on the ambient conditions (temperature and humidity).Additional peel stresses at the edges of the bond during loading amplify the effects mentioned above.


The single lap shear strength, τ_SLJ_, did not exceed a level of about 7 MPa. As those high strength samples exhibited a cohesive failure (cf. Figure 18), the interlaminar shear strength of the layer wise extruded PLA was reached. The most likely explanation for such a low interlaminar shear strength (in comparison to, e.g., [42]) is the molecular structure of the PLA generated by the ME process. Usually, after the deposition in a “standard” ME process, the polymer cools rapidly with some 10 K/s below its crystallization, T_c_, or even glass transition temperature, T_g_, (cf. e.g., [19,21]). The consequence is a mostly amorphous structure with an increased free volume and chain mobility (cf. Figure 4 heat cycle 1, which measures the polymer properties after rapid cooling). In this study, due to the heated substrate, the temperature of the polymer initially dropped to a plateau temperature, T_IR, plat_, which was above its crystallization temperature, T_c_, (cf. Figure 1, Figure 4 and Figure 9). Consequently, during the tempering and the subsequent “slow” cooling with some 10 K/min, the polymer crystallized (cf. Figure 1 and Figure 4 heat cycle 2). This resulted in a structure with an increased crystallinity and a decreased free volume and chain mobility. This led to a reduced interlaminar shear strength, as the next layer could adhere better to a previous layer with an increased free volume and chain mobility (cf., e.g., [19]).

## 4. Conclusions

Material extrusion was used to deposit d_Po_ = 0.3 mm and d_Po_ = 0.6 mm high traces of PLA on preheated aluminum substrates. Based on DSC and rheometry measurements, the substrate temperature, T_s_, was varied in between 150 and 200 °C. At those temperatures, the polymer viscosity was typical for extrusion and lay in between 10^3^ and 10^4^ Pa·s. Thermographic monitoring at an angle of 25° with respect to the substrate surface allowed a thermal evaluation of the ME process at any position on the substrate. Decreasing the layer height, d_Po_, and increasing the substrate temperature, T_s_, promoted wetting and improved the adhesion interface performance as measured in a single lap shear test. Due to thermal degradation, decreasing layer height instead of further increasing substrate temperature is favorable.

An estimation of the resulting wetting and adhesion interface performance was realized by the two parameters ${\dot{T}}_{c,avg\left(0.5-1s\right)}$ and φ_cs,i_.
${\dot{T}}_{c,avg\left(0.5-1s\right)}$ accounted for the effect of substrate temperature, T_s_. For the given processing conditions in terms of layer height, d_Po_, and substrate temperature, T_s_, the average cooling rate within 0.5 and 1 s after extrusion, ${\dot{T}}_{c,avg\left(0.5-1s\right)}$, could be used for the estimation instead of the plateau temperature, T_IR,plat_. This facilitates the evaluation during the production of an actual part, as it is difficult to monitor any point until T_IR,plat_ is reached.A variation in layer height affected the preset shape of the polymer trace in terms of φ_cs,i_ and, hence, wetting and adhesion.

For intended future work, a permanent implementation of this thermographic monitoring/estimation approach with an improved setup—a lightweight IR-camera attached to the extrusion head—offers well-defined positions for the temperature measurement. Furthermore, extensive characterization of the failure mechanisms (condition monitoring) and aging behavior of these multimaterial structures may be of interest.

## Figures and Tables

**Figure 1 materials-13-03371-f001:**
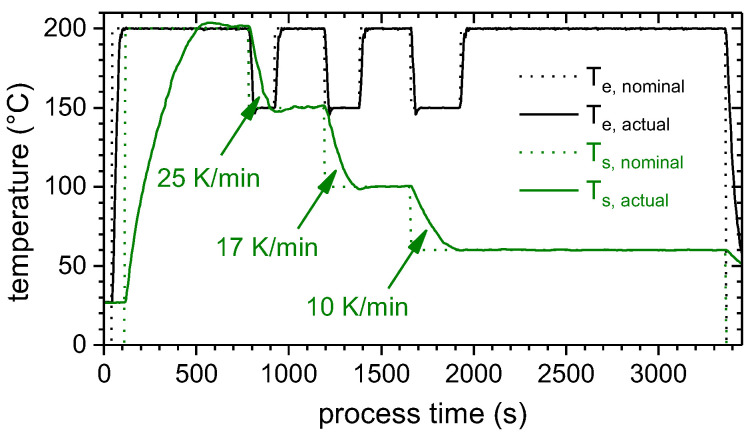
Temperature time profile of the nominal and actual temperature of the extruder and the substrate during production of a single lap joint (SLJ) specimen. The substrate temperature, T_s_, was gradually decreased after each layer. Cooling rates represent averaged values.

**Figure 2 materials-13-03371-f002:**
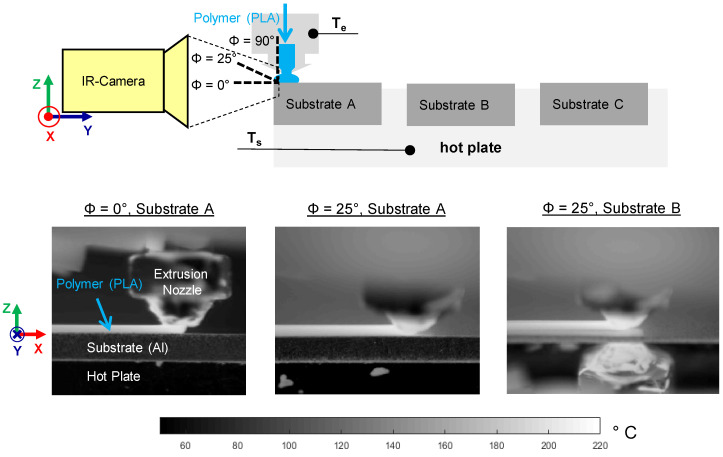
Schematic measurement setup of the three different thermographic measuring configurations with respect to the angle of vision (Φ = 0°, 25° and 90°)—top (not to scale). Extract of emissivity corrected thermograms for different angles of vision and substrate positions during the material extrusion process (T_e_ = 200 °C, T_s_ = 150 °C)—bottom.

**Figure 3 materials-13-03371-f003:**
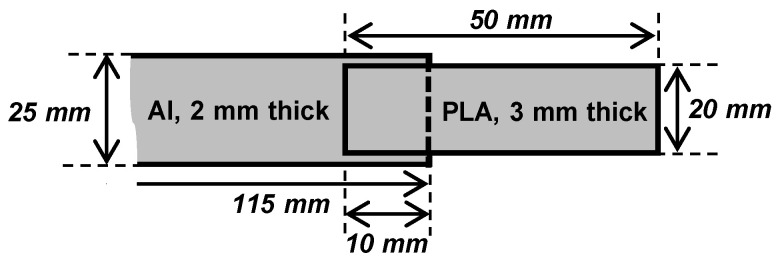
Dimensions of the single lap joint specimens (not to scale).

**Figure 4 materials-13-03371-f004:**
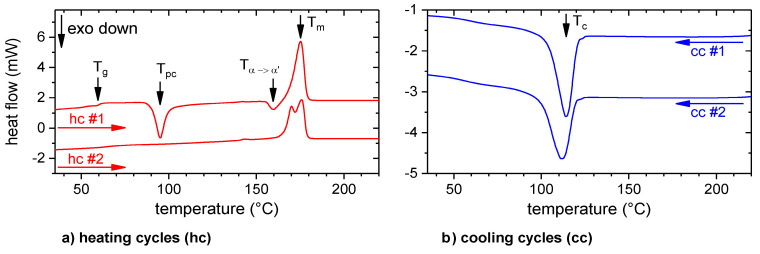
Parallel shifted heat flows of the first and second heating (hc) and cooling (cc) cycles with marked temperatures for T_g_: glass transition, T_pc_: post crystallization, T_α→α’_: crystalline phase transition, T_m_: melting and T_c_: crystallization. The heating and cooling rate was 10 K/min for each cycle.

**Figure 5 materials-13-03371-f005:**
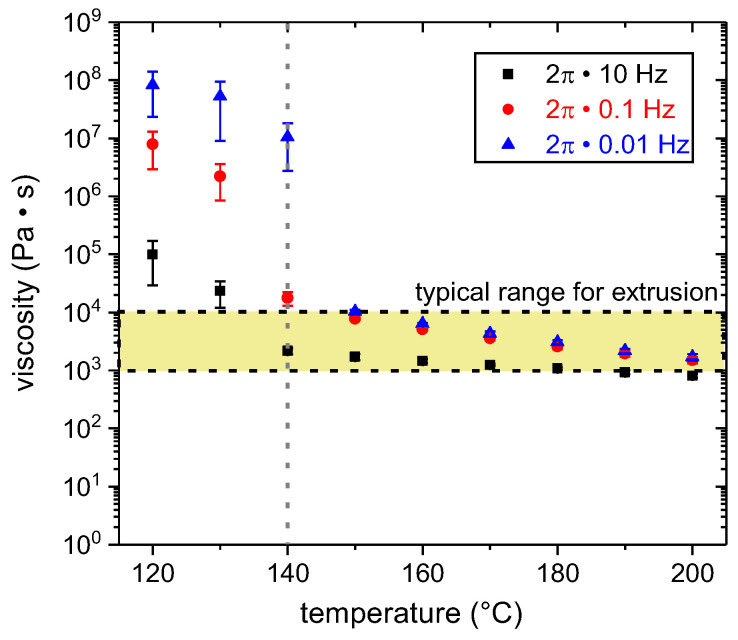
Temperature dependence of the viscosity for three different shear rates (ω = 2π· 10 Hz, 2π·0.1 Hz, 2π·0.01 Hz). The typical processing viscosity range for extrusion (10^3^–10^4^ Pa·s cf. e.g., [36]) is marked.

**Figure 6 materials-13-03371-f006:**
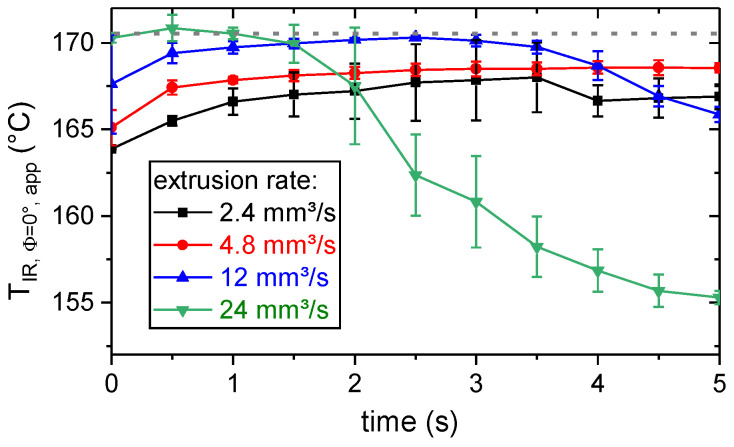
Apparent surface temperature, T_IR, Φ=0°, app_, of the PLA right below the extrusion nozzle for different extrusion rates, determined by thermography based on an assumed emissivity of 1.

**Figure 7 materials-13-03371-f007:**
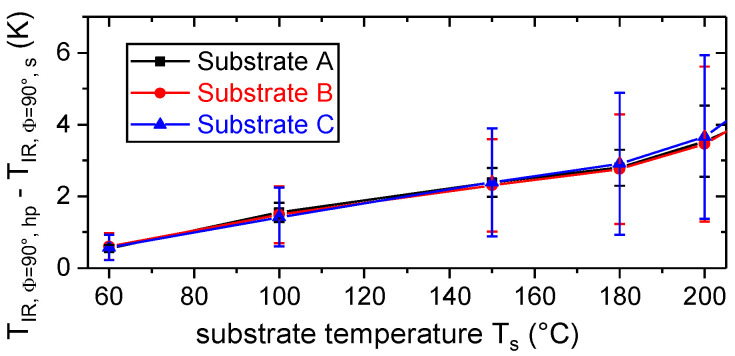
Thermographic determined difference in surface temperature between the hot plate (T_IR,Φ=90°,hp_) and the substrates (T_IR,Φ=90°,s_) mounted on the hot plate (cf. Figure 2).

**Figure 8 materials-13-03371-f008:**
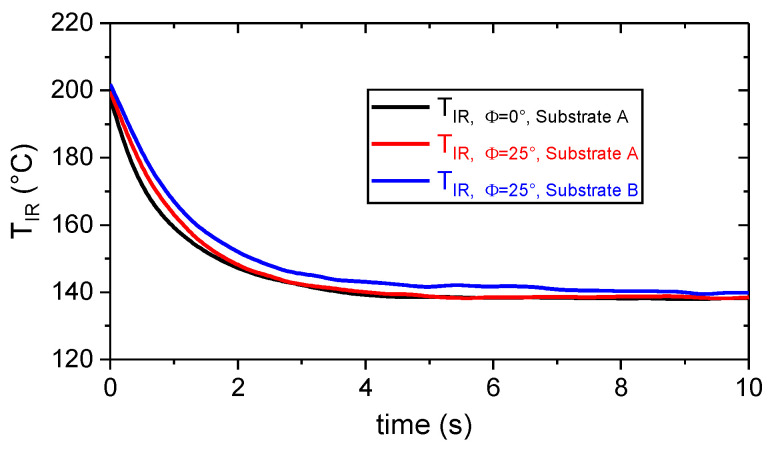
Polymer surface temperature, T_IR,_ as a function of time during the extrusion of 0.6 mm high traces on T_s_ = 150 °C hot aluminum substrates for different viewing angles, Φ, and positions on the hot plate. The temperature was determined by thermography based on an emissivity of 0.78 and the set-up presented in Figure 6.

**Figure 9 materials-13-03371-f009:**
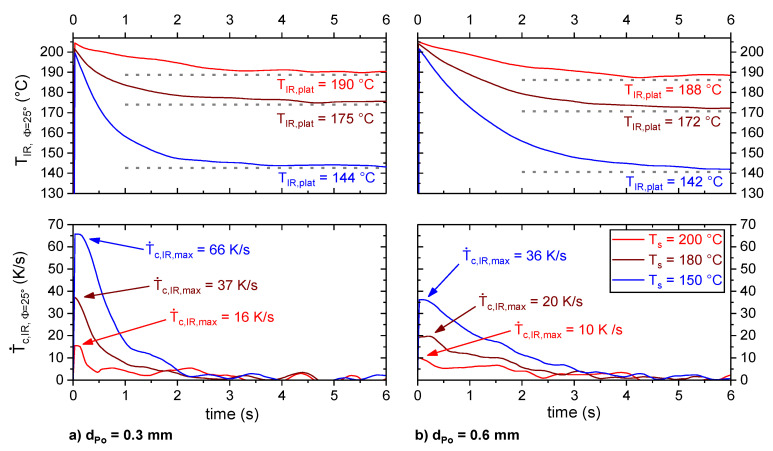
Polymer surface temperature, T_IR, Φ=25°_, as well as the corresponding cooling rate, ${\dot{T}}_{c,IR,\;\Phi =25{}^{\circ}}$, as a function of time during the extrusion of d_Po_ = 0.3 mm (**a**) and d_Po_ = 0.6 mm (**b**) high layers on T_s_ = 150 °C, 180 °C and 200 °C hot aluminum substrates. The temperature was determined by thermography based on an emissivity of 0.78 and the set-up presented in Figure 2.

**Figure 10 materials-13-03371-f010:**
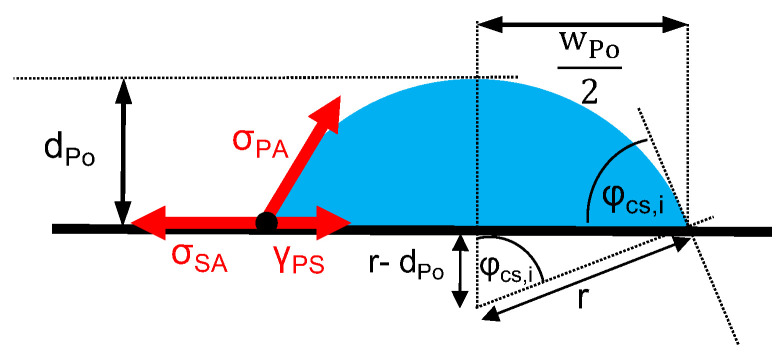
Schematic cross section of the polymer trace in the form of a cylinder segment.

**Figure 11 materials-13-03371-f011:**
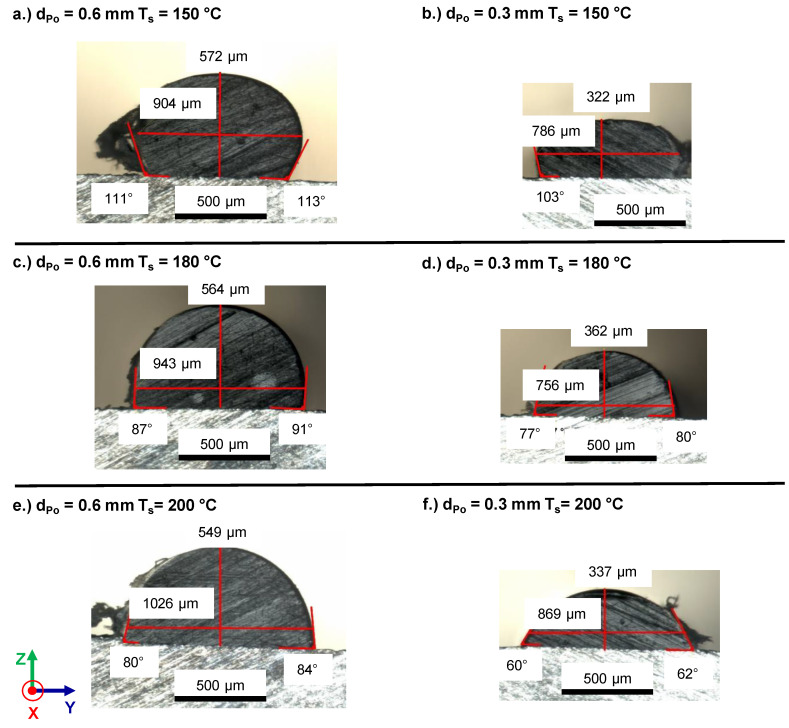
Brightfield light microscopic images for contact angle measurement for d_Po_ = 0.6 mm (**a**,**c**,**e**) and d_Po_ = 0.3 mm (**b**,**d**,**f**) high traces deposited on a substrate with a temperature of T_s_ = 150 °C (**a**,**b**), T_s_ = 180 °C (**c**,**d**) and T_s_ = 200 °C (**e**,**f**).

**Figure 12 materials-13-03371-f012:**
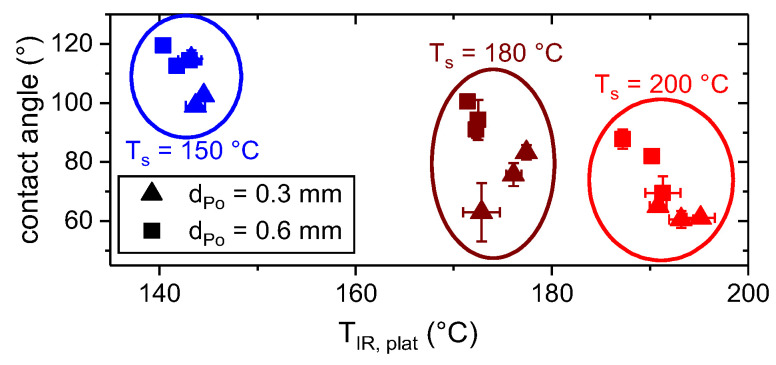
Contact angle as measured in light microscopic images (cf. Figure 11) as a function of plateau temperature, T_IR, plat_, for d_Po_ = 0.3 mm and d_Po_ = 0.6 mm high traces.

**Figure 13 materials-13-03371-f013:**
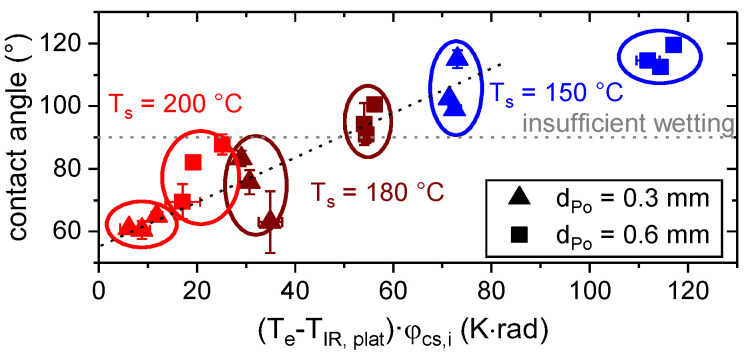
Contact angle as a function of thermographic determined plateau temperature, T_IR, plat_, and the preset contact angle, φ_cs,i_, (cf. Equation (4)).

**Figure 14 materials-13-03371-f014:**
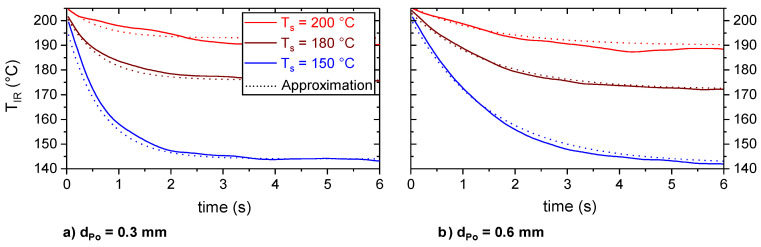
Analytical approximation according to Equation (5) of the experimental temperature profiles, T_IR_(t), for d_Po_ = 0.3 mm (**a**) and d_Po_ = 0.6 mm (**b**) high traces. For the analytical calculation, the average values of T_IR,e_ and T_IR,plat_ (cf. Table 2) were used.

**Figure 15 materials-13-03371-f015:**
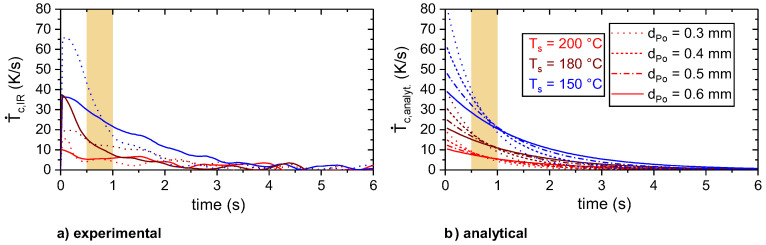
Cooling rates of the experimental (**a**) and analytical (**b**) temperature profiles for different layer heights, d_Po_, and substrate temperatures, T_s_. For the analytical approximation, T_IR,e_ and T_IR,plat_ were interpolated linearly between the average values for the layer heights 0.3 and 0.6 mm (cf. Table 2).

**Figure 16 materials-13-03371-f016:**
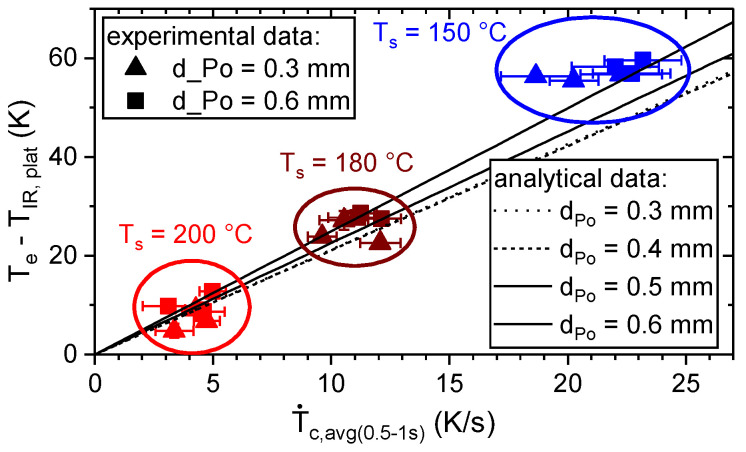
Estimation of the plateau temperature, T_IR,plat_, based on the average cooling rate ${\dot{T}}_{c,avg\left(0.5-1s\right)}$

**Figure 17 materials-13-03371-f017:**
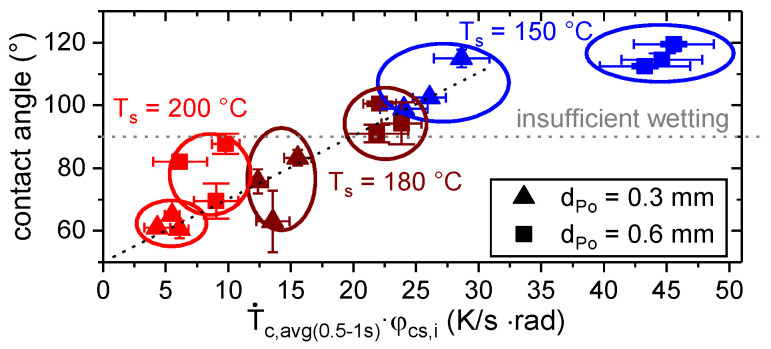
Contact angle as a function of the average cooling rate, ${\dot{T}}_{c,avg\left(0.5-1s\right)}$, and the preset contact angle, φ_cs,i_, (cf. Equation (4)).

**Figure 18 materials-13-03371-f018:**
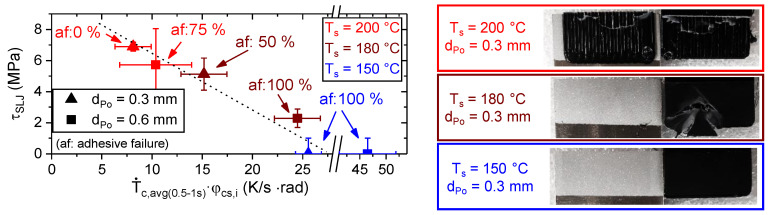
Single lap shear strength, τ_SLJ_, as a function of the average cooling rate, ${\dot{T}}_{c,avg\left(0.5-1s\right)}$, and the preset contact angle, φ_cs,i_, (cf. Equation (4)). For the calculation of the adhesive failure proportion, af, macroscopic adhesive (af: 100%), mixed (af: 50%) and cohesive (af: 0%) failure were taken into account. Exemplary fracture surfaces are presented—right.

**Table 1 materials-13-03371-t001:** Overview of selected thermal and mechanical properties of the materials used. The surface roughness of the aluminum substrates is measured with a Mahr MarTalk profilometer. Values marked with a “*” correspond to a different type of PLA.

Property		Al EN AW-6082-T6	PLA Ingeo™ 3D870
Thermal Expansion Coefficient	α (10^−6^/K)	23.1 [26]	85–185 * [31]
Thermal Conductivity	k (W/m·K)	172 [26]	0.1–0.2 * [27]
Heat Capacity	c_p_ (kJ/kg·K)	0.9 [26]	1.6–2.1 * [27]
Density	ρ (g/cm^3^)	2.71 [26]	1.07–1.25 * [27]
Melting Temperature	T_m_ (°C)	575–650 [26]	175.2 ± 0.8
Glass Transition Temperature	T_g_ (°C)		60.5 ± 0.3
Elastic Modulus	E (GPa)	70 [26]	2.9 [32]
Tensile Strength	σ_m_ (MPa)	340 [26]	40 [32]
Surface Roughness -Blank -Sandblasted (FEPA 150)	R_a_, R_z_ (µm)	0.18 ± 0.02, 1.5 ± 0.1 1.9 ± 0.5, 15 ± 4	

**Table 2 materials-13-03371-t002:** Average values for the thermographic determined extrusion, T_IR,e_, and plateau, T_IR,plat_, temperature as well as the maximum cooling rate, ${\dot{T}}_{c,IR,max}$, for different substrate temperatures (T_s_ = 200, 180, 150 °C) and layer heights (d_Po_ = 0.3, 0.6 mm).

Process Parameters		T_IR, e_ (°C)	T_IR, plat_ (°C)	${\dot{T}}_{c,IR,max}(K/s)$
T_s_ = 200 °C	d_Po_ = 0.3 mm d_Po_ = 0.6 mm	206.1 ± 0.8 205.6 ± 1.3	193.0 ± 2.2 189.6 ± 2.1	13.0 ± 2.7 8.9 ± 0.9
T_s_ = 180 °C	d_Po_ = 0.3 mm d_Po_ = 0.6 mm	203.3 ± 0.8 204.1 ± 0.6	175.5 ± 2.4 172.1 ± 0.6	38.8 ± 5.2 19.3 ± 1.6
T_s_ = 150 °C	d_Po_ = 0.3 mm d_Po_ = 0.6 mm	200.0 ± 0.4 202.1 ± 0.4	143.8 ± 0.7 141.8 ± 1.4	74.5 ± 7.4 39.1 ± 2.5

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
