# Peer review of "Estimation of the Adhesion Interface Performance in Aluminum-PLA Joints by Thermographic Monitoring of the Material Extrusion Process"

_materials, 2020, doi:10.3390/ma13153371_

Round 1

Reviewer 1 Report

The authors address an important topic in additive manufacturing. Their analysis goes in depth and is accompanied by very well prepared illustrative figures. However, some points should be altered or discussed more thoroughly.

l. 39 – carbon fibre-reinforced polymer, then an abbreviation, and consequently other abbreviations

Fig. 1 – to unify oC and K (to change K for oC) and consequently in the text

Table 1 – explanation of symbols

Fig. 5 – The experimental data for 2π x (0.01 and 0.1) are more or less identical for temperatures exceeding 140 oC. Why?

l. 289 – The emissivity of the polymer is estimated as 0.78 according to Eq. (2). Could this sentence be discussed in more detail including a linear participation of emissivity in this relation?

l. 291 – “emissivity … is considered to be independent of the temperature” –Should it be justified in more detail?

Fig. 6 – Some figures presented in oC, some in K.

Figs. 9 + 15 – Why the ordinates are not presented also for negative values?

Language – requires improvement

Conclusion: The manuscript is carefully prepared but some crucial simplifying assumptions should be justified in more detail as their introducing strongly participates in further analysis.

Reviewer 2 Report

In this work, the thermographic monitoring system was introduced to characterize the thermal process of Al-PLA joints fabricated by FDM. The influence of substrate temperature and layer height on the interfacial behavior of Al-PLA joint were investigated, and obtained the key factors affecting the wetting behavior of Al substrate. The subject of this paper is interesting and would be important for additive manufacturing. And, the manuscript is well designed and the results and conclusions presented are, most of the time, clearly explained. I think it can be acceptable for publication after address the follow minor comments.

1-Can you provide the fracture surface of sample in Fig.18?

2-Please give comments on the failure mode of single lap sample change with cooling rate and contact angle in Fig.18?

3-Did the pores in the joint interface influence the temperature results measured by thermography?

4-The thickness of Al and PLA is different, so how did you guarantee the vertically stretched during the single lap shear testing process ?
